# Real-World Efficacy and Safety of the Subcutaneous Implantable Cardioverter Defibrillator: Insights from the GASP Registry

**DOI:** 10.3390/biomedicines13071510

**Published:** 2025-06-20

**Authors:** Nikias Milaras, Evangelos Oikonomou, Konstantinos P. Letsas, Nikolaos Ktenopoulos, Sotirios Xydonas, Panagiotis Korantzopoulos, Georgios Leventopoulos, Panagiotis Dourvas, Stefanos Archontakis, Athena Batsouli, Panagiotis Mililis, Athanasios Saplaouras, Emmanuel Kanoupakis, Konstantinos Toutouzas, Stylianos Paraskevaidis, Michalis Efremidis, Skevos Sideris

**Affiliations:** 1Cardiology Department, General Hospital of Athens Ippokrateion, Vasilissis Sofias 114, 11527 Athens, Greece; panosdour92@gmail.com (P.D.); stef6arch@yahoo.com (S.A.); skevos1@otenet.gr (S.S.); 2Medical School, National and Kapodistrian University of Athens, 11527 Athens, Greece; boikono@gmail.com (E.O.); nikosktenop@gmail.com (N.K.); ktoutouz@gmail.com (K.T.); 3Department of Cardiology, Onassis Cardiac Surgery Center Athens, 17674 Athens, Greece; kletsas@gmail.com (K.P.L.); petemil@gmail.com (P.M.); saplaouras@hotmail.com (A.S.); efremidis@ocsc.gr (M.E.); 4Second Department of Cardiology, Evangelismos General Hospital of Athens, 10676 Athens, Greece; sotxyd@gmail.com (S.X.);; 5First Department of Cardiology, Faculty of Medicine, University of Ioannina, 45110 Ioannina, Greece; p.korantzopoulos@yahoo.gr; 6Department of Cardiology, University Hospital of Patras, 26504 Rion-Patras, Greece; levent2669@hotmail.com; 7Department of Cardiology, University Hospital of Heraklion, 71500 Crete, Greece; kanoup@gmail.com; 8First Cardiology Department, AHEPA University Hospital, Aristotle University of Thessaloniki, 54636 Thessaloniki, Greece; stparask@otenet.gr

**Keywords:** S-ICD, subcutaneous ICD, sudden death, defibrillator

## Abstract

**Background:** The advent of subcutaneous implantable cardioverter defibrillators (S-ICDs) marked a significant milestone in the course of cardiac rhythm devices, particularly for patients who are deemed at high risk for ventricular arrhythmias and sudden cardiac death. This extracardiac approach makes the S-ICD an especially valuable option for young patients, those with difficult venous access, or those at high risk of infection. Although the S-ICD does not provide pacing for bradycardia or heart failure, it has shown efficacy in treating ventricular arrhythmias while minimizing complications associated with transvenous systems. **Methods**: The purpose of this multicenter retrospective analysis was to assess the real-world efficacy and safety of the S-ICD in a heterogeneous population. **Results**: The GASP registry consisted of 114 patients, 68% male, aged 41 ± 15 years, with a mean LVEF of 50%. In the follow-up of 35 months, inappropriate shocks occurred in 7% while appropriate shocks occurred in 6.2%. The most common reasons for inappropriate shocks were myopotentials and atrial tachyarrhythmias. Thirty-day complication-free rates were 97.3%, with the majority of patients requiring device extraction due to infection. Over the longer term, four patients required re-intervention due to local discomfort, while one device was extracted for infection. In a multivariate analysis, complications were not significantly higher in the sicker population, such as those with diabetes, kidney disease requiring dialysis, or heart failure. **Conclusions**: These findings support the growing role of the S-ICD as an alternative to the TV-ICD, especially in patients without pacing indications.

## 1. Introduction

It has been almost 50 years since the first transvenous implantable cardioverter defibrillator (TV-ICD) was introduced for commercial use, and registries indicate a marked reduction in sudden cardiac death in patients with heart failure [1]. Since then, evolution in pharmacotherapy reduced mortality and symptoms in those patients substantially. Despite their high efficacy, TV-ICDs are encircled with a number of adverse events such as lead failure (almost 20% in 10 years) [2], infection, and tricuspid valve insufficiency, reducing the overall efficacy of the device and impacting patient quality of life and health-related costs. An appealing alternative to the transvenous ICD is the subcutaneous ICD (S-ICD). It consists of an extracardiac lead positioned in the subcutaneous fat in front of the sternum and a battery-generator usually positioned in the left mid-axillary line, either subcutaneously or intermuscularly.

However, the S-ICD has certain limitations compared to the TV-ICD in specific aspects of device performance. Currently, a defibrillation test (DFT) is recommended after S-ICD implantation to confirm device efficacy in terminating induced ventricular fibrillation (VF) [3,4]. Although the PRAETORIAN score has been shown to be reliable in predicting successful DFT using plain chest radiography, it can only be utilized after the device has already been implanted [4].

Perhaps the biggest shortcoming of the S-ICD is the lack of pacing capabilities. This has dual implications: (a) anti-tachycardia pacing (ATP) is not possible. Diseases with re-entrant, scar-related ventricular tachycardia (VT), such as ischemic and dilated cardiomyopathy, may benefit from ATP; 72% of VTs were successfully treated with ATP regardless of tachycardia cycle length, leading to a better quality of life in patients with an ICD [5]. This makes the S-ICD a better choice for patients with primary “electrical disease”, where spontaneous ventricular fibrillation is better treated with defibrillation. (b) Bradycardia pacing is also not possible with the S-ICD, ruling out older patients with bystanding degeneration of the heart’s conduction system or younger patients with genetic mutations necessitating pacing. Recently, a new combination of a leadless pacemaker with an S-ICD was tested, with promising results that are also capable of ATP [6].

Inappropriate shocks are also among the caveats of S-ICD usage. P or T wave oversensing regularly led to defibrillation in the older generation devices, with triple rates of IAS when compared to the traditional TV-ICD. Nevertheless, software updates such as the SMART pass filter and dual-zone programming led to a 4–5% of IAS at one-year follow-up, which approximates that of the TV-ICD [7]. In the present study, we present data on S-ICD implantation in five centers in Greece between 2016 and 2024.

## 2. Methods

The current study is a retrospective, multicenter observational study of patients in whom an S-ICD was implanted as standard-of-care in Greece. Patients were treated according to current guidelines, and no additional treatment was imposed for study purposes. Data extracted from the medical records were processed anonymously. Therefore, in accordance with Greek law, the Institutional Review Board waived the need for written informed consent.

### 2.1. Study Population

This cohort consists of 114 subjects who were implanted with an S-ICD in five hospitals in Greece from October 2015 up to July 2024. Follow-up data were recorded in August 2024. Somatometric features, cardiomyopathy substrate, peri-implantation data, and current follow-up were recorded in order to outline the challenges and adverse effects of the S-ICD in a real-world setting. The subjects were followed up for a median time of 35 months (IQR 14, 60).

### 2.2. Definitions

All somatometric, ECG, and echocardiographic indices were collected at the time of implantation. Kidney disease was defined as the need for hemodialysis, diabetes was defined according to the latest American Diabetes Association criteria, and appropriate shocks were defined as S-ICD shocks for VT or VF above the preconceived rate (as chosen by the implanter), while all other shocks were declared inappropriate. Procedural techniques were left to the operator’s discretion according to their standard practices, including conversion testing, surgical technique, and anesthesia. S-ICD system- and procedure-related complications were defined as complications that were caused by or would not have occurred in the absence of the S-ICD. All complications after 30 days post-implantation were considered late.

## 3. Statistical Analysis

The normality of distribution was tested with the Kolmogorov–Smirnov test and visual inspection of P-P plots. Data were presented as mean ± standard deviation (SD) for continuous variables following a normal distribution. Categorical variables were summarized as counts and percentages. For comparisons between groups, the Student’s *t*-test was used for continuous variables, depending on the normality of data distribution. Analysis of Variance was used to test for intergroup differences in continuous variables when three or more group categories were present. Categorical variables were compared using the chi-square or Fisher’s exact test, as appropriate. Two-sided hypotheses were used for the analysis. Statistical significance was considered when the *p*-value was lower than 0.05. All statistical calculations were performed in SPSS software (version 27.0, SPSS, Inc. IBM Corp., Armonk, NY, USA).

## 4. Results

### 4.1. Study Population Characteristics

The GASP (Greek Analysis of the Subcutaneous-ICD Practice) registry consisted of 114 patients aged 41 ± 15 years, reflecting that S-ICDs are usually reserved for younger patients, with the majority of them being male (68%) (Table 1). The mean BMI of the population was 26.11 ± 5.14 kg/m^2^. The underlying cardiomyopathy diagnoses were diverse, with the most common being dilated cardiomyopathy (22.8%), hypertrophic cardiomyopathy (21.1%), and ischemic cardiomyopathy (20.2%) (Figure 1 left panel). S-ICD was implanted in most of the cases on the basis of a primary prevention strategy (82.5%). Most of the patients had an LVEF ≥ 50% (53.5%), and 26.3% had an LVEF ≤ 40% (Figure 1 right panel). Reflecting current guidelines, NYHA class III and II were evident in 8.8% and 30.7% of the study’s population, respectively, while there were no subjects in NYHA class IV. The mean QRS duration was 97 msec, and 10.7% of the subjects had permanent atrial fibrillation.

### 4.2. Implantation Details

As per current manufacturer recommendations, patients underwent ECG screening to assess compatibility with S-ICD sensing. A passing vector is a sensing configuration that provides clear, stable ECG signals with an adequate QRS-to-T-wave ratio across different postures, ensuring reliable arrhythmia detection without oversensing. At least one of three vectors in the supine and standing positions was required to pass the screening test. Patients passing at least two and three ECG screening vectors were 94.7% (Figure 2 upper left panel). The primary vector (inferior electrode to can) was the most commonly suitable vector. The proportions of three passing vectors in each cardiomyopathy are shown in Table 2, and there was no difference according to cardiomyopathy type (*p* = 0.308).

Regarding the use of passing vectors, there was no difference in BMI according to the use of one or two passing vectors compared to the group of patients with three passing vectors (25.52 ± 3.84 kg/m^2^ vs. 26.53 ± 5.88 kg/m^2^, *p* = 0.31).

Notably, subjects with a risk of conversion failure according to PRAETORIAN score have increased BMI when compared to patients with intermediate and low risk of conversion failure (24.69 ± 4.58 kg/m^2^ vs. 24.81 ± 3.56 kg/m^2^ vs. 33.93 ± 9.87 kg/m^2^, *p* < 0.001) (Figure 3).

Since the first S-ICD was implanted in 2015, the two-incision technique was used in 89.5% of the cases (Figure 2 upper middle panel), and the SMART pass filter was active in 96.5% (Figure 2 upper right panel). Also, general anesthesia was utilized in all of the procedures.

Defibrillation testing (DFT) was conducted in all patients regardless of the PRAETORIAN score. VT/VF induction was possible in all patients. At least 65 J of energy was used for DFT up to a maximum of 80 J, achieving conversion to sinus rhythm in all but one patient (Figure 2 lower left panel). Of note, in the one patient where DFT failed, external biphasic cardioversion of 270 J also failed to convert VF to sinus rhythm four times, and the patient spontaneously reverted to sinus rhythm.

PRAETORIAN score analysis was possible in 58 patients in whom both anteroposterior and lateral chest X-rays were obtained. According to the PRAETORIAN trial [8], scores between 30 and 90 were considered a marker of low risk of DFT failure, 90 to 150 were intermediate risk, and scores above 150 were considered high risk. The low-risk group comprised 64.1%, the intermediate risk 26.6%, and the high-risk group 9.4% (Figure 2 lower right panel). Our only patient with DFT failure had a PRAETORIAN score of 150.

### 4.3. Complications

Device- and procedure-related complications that occurred within 30 days of implantation are listed in Table 2. Freedom from complications directly caused by the S-ICD device at 30 days was 97.3%. Two S-ICDs were explanted in the first month post-implantation due to pocket infection. Notably, in both patients, the device was implanted after extraction of a previous transvenous ICD due to infection. The third case was lead repositioning due to translocation shortly after implantation.

All complications after 30 days were considered late (5.7%). In most of the cases (4), re-intervention was needed due to local discomfort; there was one late infection with concomitant device extraction and one extraction in the patient with failed DFT. Regarding the incidence of 30-day complications, over-30-day complications, device extraction, and re-interventions, there was no difference according to patients’ BMI. Battery replacement was performed on 11 patients (9.7%).

Similarly, there was no difference in the incidence of complications according to chronic kidney disease.

### 4.4. Patients’ Prognosis

During the follow-up time of 35 months, five patients (4.4%) died. Three died from non-cardiovascular causes, one due to heart failure exacerbation, and one patient died suddenly. Fifteen of our patients have been hospitalized (13.2%). Of those, six (40%) have been hospitalized for heart failure decompensation, three (20%) for VT episodes or electrical storm, three (20%) for non-ICD related infection, one (7%) for lead repositioning, and two (13%) for ICD infection.

### 4.5. Shocks

Inappropriate shocks (IASs) were evident in eight patients (7%) and appropriate shocks in seven cases (6.2%). One patient had both inappropriate and appropriate shocks. None of the patients with shocks died over the follow-up time. Four patients experienced IAS due to myopotentials, three due to atrial fibrillation or atrial tachycardia, and one due to P wave oversensing. As shown in Table 3, there was no difference in appropriate and inappropriate shocks according to the incision procedural technique.

In a multivariate analysis, there was no difference in the incidence of IAS according to sex, BMI, height, and PRAETORIAN score in our registry. Interestingly, in subjects with primary arrhythmogenic syndrome and congenital heart disease, the incidence of inappropriate shocks increased (Table 4).

## 5. Discussion

The results of this study contribute to the growing body of evidence supporting the use of S-ICDs as an alternative to transvenous ICDs in patients at risk of sudden cardiac death. Our findings align with previous studies showing that S-ICDs offer comparable efficacy in arrhythmia detection and defibrillation while presenting fewer long-term complications associated with transvenous leads, such as lead fractures, venous obstruction, and infections (graphical abstract).

This study’s population is somewhat different when compared to the literature for two reasons. The average age of the cohort, 41 ± 15 years, underscores the increasing preference for S-ICDs in younger patients who are at high risk for sudden cardiac death but may benefit from avoiding the long-term complications associated with transvenous leads. Secondly, our findings indicate that S-ICDs were predominantly implanted for primary prevention, with a majority of patients demonstrating preserved or mildly reduced LVEF. This contrasts with traditional ICD populations, where secondary prevention and severely reduced LVEF are often the primary indications.

In the present study, the incidence of IAS (7%) was not disproportionately different and comparable with published data from the EFFORTLESS registry [9,10]. In the EFFORTLESS multicenter registry, 8.3% of 581 patients experienced inappropriate shocks. IASs are also similarly prevalent in patients treated with the traditional transvenous ICD [11]. In this cohort of 1544 patients, 13% experienced IASs, with each shock increasing the all-cause mortality up to a hazard ratio of 3.7. Even though IASs are more probable in patients with Brugada syndrome or HCM due to T wave oversensing, they did not reach statistical significance in our population [12].

On the contrary, appropriate shocks were low (6.2%) when compared to the literature. In a pooled analysis of the IDE and EFFORTLESS trials, 10.5% of patients received at least one appropriate shock over a 3-year follow-up [13]. Patients in this study were slightly older at 52 vs. 41 years in our registry, and the mean EF was significantly lower at 39.4 vs. 50%. All of our patients presented with an indication for ICD implantation according to the current ESC guidelines [14], but the low shock rates may be attributed to the following: (a) most of our patients received the S-ICD in a primary prevention basis, (b) the relatively preserved EF of our population is generally associated with less ventricular tachyarrhythmias, and (c) the relatively short follow-up of 35 months might not be enough to fully elicit appropriate shocks in this young population.

Apropos the passing vectors, there was no difference in BMI according to the use of one or two passing vectors compared to the group of patients with three passing vectors. This was also independent of cardiomyopathy type, even though Brugada syndrome and HCM patients are reportedly more likely to fail pre-implantation screening for the S-ICD [15,16].

The primary vector (proximal electrode ring to can) was the most commonly utilized vector, and this comes in accordance with IDE and EFFORTLESS [9,10]. These data support the current use of the primary sensing vector as the nominal vector for the S-ICD system.

The PRAETORIAN score is a relatively simple scoring system that is calculated after S-ICD implantation from an anteroposterior and lateral chest X-ray. In simple terms, the longer the distance of the S-ICD can from the thoracic cage or the S-ICD lead from the sternum, the higher the PRAETORIAN score and the lower the chance of VF conversion to sinus rhythm. In our study, this score was calculated in a subset of patients with available chest X-rays, and what makes no impression is the fact that the higher the BMI of the patients, the higher the score. Recently, the PRAETORIAN–DFT trial randomized patients in a 1:1 fashion to DFT or no DFT according to their PRAETORIAN score [4]. A score of <90 had a negative predictive value of 99% for successful DFT. With growing experience and submuscular can implantation, there will be no need for DFT except maybe in a subset of patients, such as those with high BMI.

Overall, the GASP complication-free event rate at 30 days was 97.3%, which is consistent with the PAS trial [17]. In a multivariate analysis, 30-day complications were not significantly higher in the sicker population, such as those with diabetes, kidney disease requiring dialysis, or heart failure. This is of importance since complications such as ICD infection in this subset of patients bear greater mortality.

With regards to late complications, over the follow-up period of 35 months, local discomfort was reported as the main reason for re-intervention in four patients and was performed in one center. The other two were due to infection and failed DFT post-implantation. This highlights the fact that most complications arise in the first 30 days after surgery, and complication rates are generally low. Results from studies with longer follow-ups demonstrate no structural lead failures at 5 [18] or 10 [19] years, while pocket infection rates are generally low, but studies with longer follow-up are needed [20].

## 6. Limitations

The present study should be interpreted in light of certain methodological limitations. The chief limitation is the retrospective nature of the analysis. Furthermore, inclusion into this study required screening success and an indication for ICD implantation according to currently published ESC guidelines [14]. Other criteria, such as prior transvenous ICD infection, were not recorded in all patients for the purposes of the GASP registry. In addition, programming and intraoperative management of patients, such as intramuscular can placement, was not prescriptive. Finally, zone programming and events below the shock zone that did not require therapy were not recorded. This allows for real-world clinical use of the S-ICD but may increase variability in outcomes between centers.

## 7. Conclusions

In this real-world, Greek multicenter registry, the S-ICD demonstrated favorable efficacy and safety profiles in a diverse population of relatively young patients, predominantly implanted for primary prevention. While the absence of pacing capabilities remains a limitation, recent advances in device technology and implantation strategies—such as intramuscular positioning and integration with leadless pacing systems—may further broaden the applicability of the S-ICD. Larger, prospective studies with longer follow-ups and direct comparisons to the TV-ICD are warranted to evaluate long-term outcomes, particularly regarding late complications, lead durability, and device longevity.

## Figures and Tables

**Figure 1 biomedicines-13-01510-f001:**
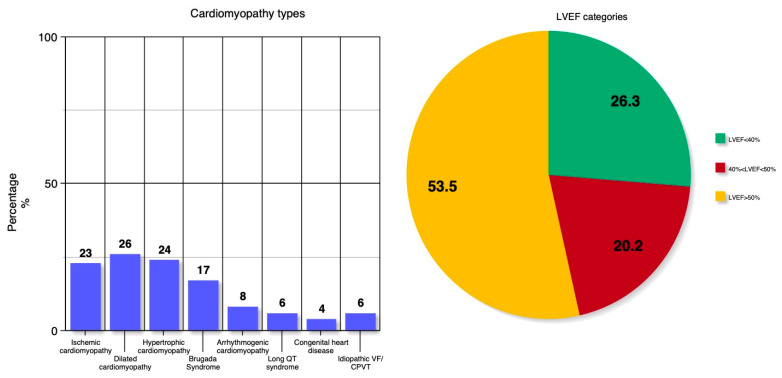
(**Left**) panel: Bar heights represent valid percentages of cardiomyopathy/channelopathy of the study population. (**Right**) panel: Valid percentages of the patient’s categorization according to LVEF categories LVEF ≤ 40%, LVEF > 40% and <50%, and LVEF ≥ 50%. VF: ventricular fibrillation; CPVT: catecholaminergic polymorphic ventricular tachycardia; LVEF: left ventricular ejection fraction.

**Figure 2 biomedicines-13-01510-f002:**
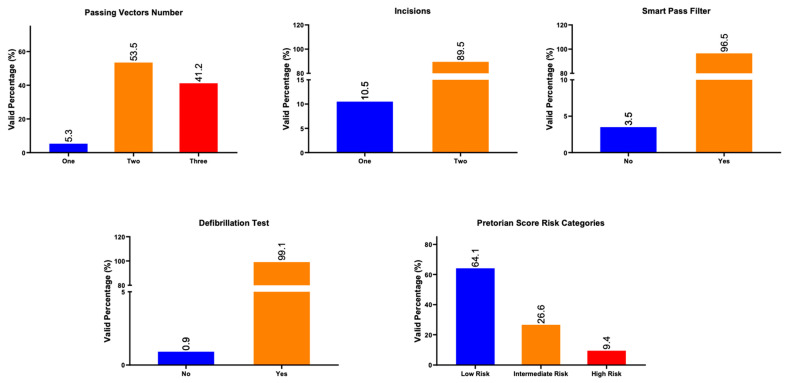
Upper left panel: Bar heights represent valid percentages per passing vector used. Upper middle panel: Bar heights represent valid percentages of the number of incision techniques used. Upper right panel: Bar heights represent valid percentages of the SMART pass filter technique used. Lower left panel: Bar heights represent valid percentages of defibrillation tests used. Lower right panel: Bar heights represent valid percentages of risk of conversion failure according to PRAETORIAN score categories.

**Figure 3 biomedicines-13-01510-f003:**
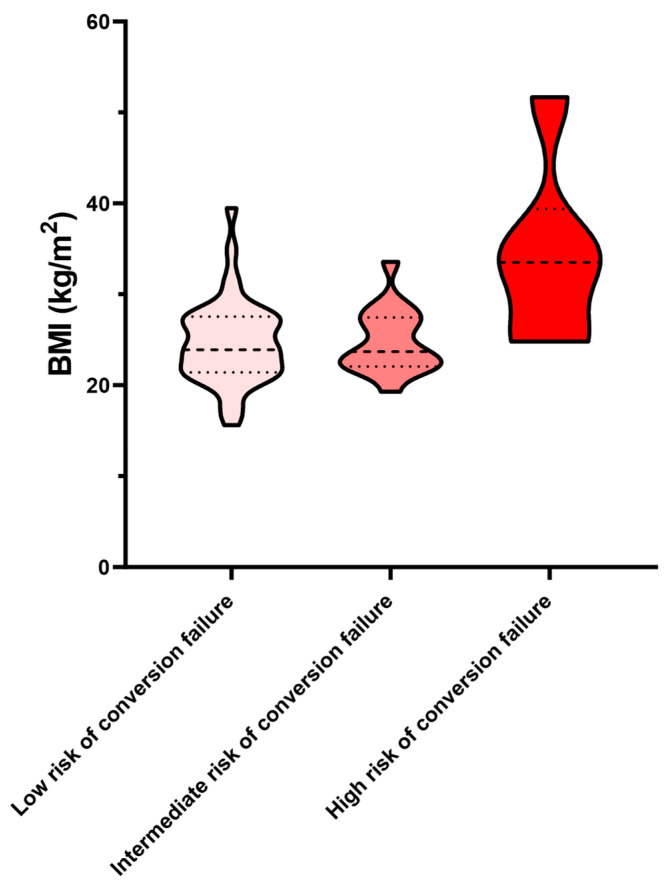
Violin plot showing the distribution of BMI across PRAETORIAN score categories. The width of each “violin” represents the probability density of the data at different values, with wider sections indicating higher density. The thick dashed line within each violin marks the median, and the smoother dashed lines indicate the interquartile range (IQR).

**Table 1 biomedicines-13-01510-t001:** Demographic and clinical characteristics of the study population.

Number of Subjects	114
Age (Years)	41 ± 15
Male Sex (%; n)	68.4; 78
Body Mass Index (kg/m^2^)	26.11 ± 5.14
Type of Cardiomyopathy/Channelopathy	
Ischemic Cardiomyopathy (%; n)	20.2; 23
Dilated Cardiomyopathy (%; n)	22.8; 26
Hypertrophic Cardiomyopathy (%; n)	21.1; 24
Brugada Syndrome (%; n)	14.9; 17
Arrhythmogenic Cardiomyopathy (%; n)	7.0; 8
Long QT Syndrome (%; n)	5.3; 6
Congenital Heart Disease (%; n)	3.5; 4
Other (Idiopathic VT, catecholaminergic	
polymorphic VT, Mitral Annular Dysjunction, etc.)	5.3; 6
Primary Prevention (%; n)	82.5; 94
QRS Duration (msec)	97 ± 17
Atrial Fibrillation (%; n)	10.7; 12
Left Ventricular Ejection Fraction (%)	50 (39, 60)
LVEF Categories	
LVEF ≤ 40% (%; n)	26.3; 30
LVEF > 40% and <50% (%; n)	20.2; 23
LVEF ≥ 50% (%; n)	53.5; 61
New York Heart Association Functional Classification	
I (%; n)	60.5; 69
II (%; n)	30.7; 35
III (%; n)	8.8; 10
Diabetes Mellitus (%; n)	8.8, 10
Chronic Kidney Disease	5.3; 6

n: number of subjects; VT ventricular tachycardia; LVEF: left ventricular ejection fraction.

**Table 2 biomedicines-13-01510-t002:** Implantation details.

Passing Vector Use (%; n)	
One (%; n)	5.3; 6
Two (%; n)	53.5; 61
Three (%; n)	41.2; 47
Two-Incision Technique (%; n)	89.5; 102
SMART Pass Filter (%; n)	96.5; 109
DFT Test Successful (%; n)	99.1; 113
PRAETORIAN Score	
Low Risk of Conversion Failure (%; n)	64.1; 41
Intermediate Risk of Conversion Failure (%; n)	26.6; 17
High Risk of Conversion Failure (%; n)	9.4; 6
30-Day Complications (%; n)	2.7; 3
Over-30-Day Complications (%; n)	6.1; 7
Re-intervention (%; n)	5.3; 6
Device Extraction (%; n)	3.5; 4
Battery Replacement (%; n)	9.7; 11
Three Passing Vectors	
ICM	56.5; 13
DCM	38.5; 10
HCM	45.8; 11
Brugada	23.5; 4
ACM	37.5; 3
LQT	16.7; 1
Congenital Heart Disease	25.0; 1
Other	66.7; 4

Values are present as valid percentages. DFT: defibrillation test; n: number of subjects; ICM: ischemic cardiomyopathy; DCM: dilated cardiomyopathy; HCM: hypertrophic cardiomyopathy; ACM: arrhythmogenic cardiomyopathy; LQT: long QT syndrome.

**Table 3 biomedicines-13-01510-t003:** Incidence of inappropriate and appropriate shocks according to incision technique.

	Inappropriate Shock			*p*-Value	Appropriate Shock	*p*-Value
	No	Yes		No	Yes	
Passing Vector Use (%)						
One	100.0	0.0		100	0.0	
Two	91.8	8.2	0.73	90.2	9.8	0.21
Three	93.6	6.4		97.8	2.2	
Two-Incision Technique (%)						
No Yes	83.8 94.1	16.7 5.9	0.16	91.7 94.1	8.3 5.9	0.75
SMART Pass Filter (%)						
No Yes	75.0 93.6	3.5 96.5	0.16	100.0 94.4	0.0 5.4	0.63
DFT Test (%)						
No	100.0	0.0		100.0	0.0	
			0.78			0.80
Yes	92.9	7.1		93.8	6.2	
PRAETORIAN Score						
Low Risk of Conversion Failure	92.7	75		62.5	2.4	
Intermediate Risk of Conversion Failure	100.0	0.0	0.31	25.0	5.9	0.71
High Risk of Conversion Failure	83.3	16.7		9.4	0.0	

DFT: defibrillation test.

**Table 4 biomedicines-13-01510-t004:** Inappropriate shock incidences according to patient characteristics.

	Inappropriate Shocks		*p*-Value
	Yes	No	
Male Gender (%; n)	6.1; 7	62.3; 71	0.22
BMI (kg/m^2^)	28.39 ± 3.47	25.96 ± 5.22	0.20
Height (cm)	174 ± 9	173 ± 6	0.68
PRAETORIAN Score			
Low Risk of Conversion Failure (%; n)	7.3; 3	92.7; 38	
Intermediate Risk of Conversion Failure (%; n)	0.0; 0	100.0; 17	0.31
High Risk of Conversion Failure (%; n)	16.7; 1	83.3; 5	
Cardiomyopathy Type			
ICM	0.0; 0	100.0; 23	
DCM	0.0; 0	100.0; 26	
HCM	4.2; 1	95.8;23	0.05
ACM	12.5; 1	87.5; 7	
Primary Arrhythmogenic Syndrome	17.4; 4	82.6; 19	
Congenital Heart Disease	25.0;1	75.0; 3	

BMI: body mass index; ICM: Iichemic cardiomyopathy; DCM: dilated cardiomyopathy; HCM: hypertrophic cardiomyopathy; ACM: arrhythmogenic cardiomyopathy.

## Data Availability

Data are available upon request from the corresponding author.

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
