# Peer review of "Real-World Efficacy and Safety of the Subcutaneous Implantable Cardioverter Defibrillator: Insights from the GASP Registry"

_biomedicines, 2025, doi:10.3390/biomedicines13071510_

Round 1

Reviewer 1 Report

Comments and Suggestions for Authors

This is an interesting, sound report, generally well-written. There are a few awkward usages and minor grammatical issues. I include a marked-up copy of the manuscript documenting these. Only minor changes are recommended.

Comments on the Quality of English Language

The text is quite readable. There are scattered minor issues that I recommend addressing, shown in the attachment.

Author Response

We really thank the reviewer for such a comprehensive review of our manuscript. Most of the issues raised are now corrected (including grammatical errors, citations and graphical abstract issues). Changes are marked in yellow.

Numbers in institutional affiliations are now superscripted.

The Graphical abstract is now mentioned in the discussion section.

I (the corresponding author) did not know that one could not begin a sentence with a number. This is now fixed throughout the text. Thank you for pointing this out.

Muscle artifacts=myopotentials . This is a term commonly used in implantable device literature.

The first Figure in the graphical abstract is from boston scientific and it requires this subheading for the use. You are completely right though it seems as is the company has something to do with this manuscript. The figure is now changed with a non copyrighted one. Furthermore, heading of figures and the figures themselves are now bigger for more clarity.

Reviewer 2 Report

Comments and Suggestions for Authors

The authors present a multicenter retrospective study from the GASP registry, evaluating the real-world safety and efficacy of the subcutaneous implantable cardioverter defibrillator (S-ICD) across five centers in Greece between 2016 and 2024. The study includes 114 patients with a mean age of 41 years and a mean follow-up of 35 months. The data show a 7% rate of inappropriate shocks and a 6.2% rate of appropriate therapies, with the most common causes of inappropriate shocks being myopotentials and atrial tachyarrhythmias. The short-term complication-free rate was high (97.3%), and long-term complications were limited. 

The study provides important real-world insights into S-ICD performance, especially in a younger population with relatively preserved ejection fraction. However, there are several limitations and areas requiring clarification or improvement.

Firstly, the conclusion presented in the abstract—that the S-ICD is a "worthy alternative" to transvenous ICDs—is not fully supported by the data, as there was no direct comparison with transvenous ICDs. Without a comparator group, this statement appears speculative. The authors should consider modifying the conclusion to reflect the descriptive nature of the findings or, ideally, include a matched case-control cohort of patients with transvenous ICDs implanted during the same period for more robust comparative analysis.

In the introduction, the sentence “Unfortunately, the S-ICD is inferior to the TV-ICD in other aspects” should be revised for tone and scientific accuracy to a more objective phrasing 

In Section 4.1, the description of patient characteristics should be made more concise and scientific. Phrases such as “trailing behind in equal shares” should be avoided in favor of neutral and precise language 

The manuscript lacks a formal conclusion section. The authors should add a dedicated conclusion section to strengthen the manuscript.

While the tables are clear and the references are adequate, the clinical utility of the paper could be enhanced by including examples of inappropriate therapy tracings—especially for those due to myopotentials or atrial arrhythmias. This would not only provide illustrative insight into the diagnostic challenges but also help guide future optimization of device programming. Additionally, a brief discussion on strategies to minimize inappropriate shocks, including software algorithms or patient selection criteria, would be beneficial. 

Finally, given the relatively young average age of the cohort, it would be helpful to briefly comment on long-term considerations such as device longevity, potential for future leadless pacing integration, and patient satisfaction or quality of life outcomes if available.

Author Response

We thank the reviewer for his thorough review of our manuscript. (changes are marked in red)

1: You are absolutely right about this. We were overwhelmed and extrapolated our results. The S-ICD might be a worthy alternative.

2: The phrase is changed to 

“However, the S-ICD has certain limitations compared to the TV-ICD in specific aspects of device performance.”

3: This is now changed : The underlying cardiomyopathy diagnoses were diverse, with the most common being dilated cardiomyopathy (22.8%), hypertrophic cardiomyopathy (21.1%), and ischemic cardiomyopathy (20.2%) 

4: Unfortunately, we kept no electrograms of inappropriate shocks and cannot add a figure, even though this would be a great visual addition to our manuscript

5: A discussion section is now added addresing  both future perspectives and the need for studies.

Reviewer 3 Report

Comments and Suggestions for Authors

As we know the main disadvantage of TV-ICD is TR which can be very dangerous. Therefore, S-ICD is the best alternative. However, the authors should give more detail between differences and mid- and long-term follow-up.

Author Response

Thank you for your comments: when this article was written, longer term results where not available besides the EFFORTLESS study. Another study with a 10 year follow up is now available and is cited in the discussion section. (green colour)

Reviewer 4 Report

Comments and Suggestions for Authors

Real-World Efficacy and Safety of the Subcutaneous Implantable Cardioverter Defibrillator: Insights from the GASP Registry

This multicenter retrospective study from Greece evaluated the efficacy and safety of subcutaneous implantable cardioverter defibrillators (S-ICDs) in a diverse patient population over a median follow-up of 35 months. The results demonstrated that the S-ICD effectively prevented sudden cardiac death with low complication rates—97.3% of patients experienced no major complications at 30 days, and late complications such as local discomfort and infections were infrequent. Inappropriate shocks occurred in 7% of patients, primarily due to myopotentials and atrial arrhythmias, while appropriate shocks for ventricular arrhythmias occurred in 6.2%. Importantly, the device showed comparable performance in younger patients with preserved or mildly reduced systolic function, and complications did not significantly increase among higher-risk subgroups such as diabetics or those with kidney disease. Overall, the study supports S-ICDs as a safe and effective alternative to traditional transvenous ICDs, particularly for patients at high risk of device-related infections or venous access issues, although limitations related to its retrospective design and lack of standardized programming are acknowledged.

  • The manuscript would benefit from a discussion acknowledging the need for direct comparisons between S-ICDs and transvenous ICDs. I recommend the authors incorporate a suggestion for future research that includes comparative studies—either within the same cohort or using matched controls—to better evaluate differences in complication rates, quality of life, and cost-effectiveness
  • I recommend that future research include longer follow-up periods. Consistent, long-term monitoring for late-onset complications—such as lead fractures and device malfunctions—would yield important insights into the durability and long-term safety of S-ICDs, which are critical for guiding clinical decision-making.
  • I recommend that the authors include a concluding section to summarize their key findings and highlight the clinical implications of their work.

Author Response

Thank you for your insightful comments. A discussion section is now provided that includes the need for longer follow up and direct comparison with the TV-ICD.

Reviewer 5 Report

Comments and Suggestions for Authors

This is a pertinent pilot study with standard methodology. While there is no obvious shortcomings of the study, low rates of appropriate shocks do raise some concerns. Was patient population well understood? I would suggest this result is discussed in much more detail.

Author Response

Thank you for your kind comment. The shock rates are indeed low and we tried further explaining it in the discussion section (mauve colour): All of our patients presented with an indication for ICD implantation according to the current ESC guidelines but the low shock rates may be attributed to the following: a) most of our patients received the S-ICD in a primary prevention basis, b) the relatively preserved EF of our population is generally associated with less ventricular tachyarrhythmias, c) the relative short follow up of 35 months might not be enough to fully elicit appropriate shocks in this young population.